# Estimating the effect of health assessments on mortality, physical functioning and health care utilisation for women aged 75 years and older

Xenia Dolja-Gore[1], Julie E. Byles[1]*, Meredith A. Tavener[1], Catherine L. Chojenta[1], Tazeen Majeed[1], Balakrishnan R. Nair[1,2], Gita D. Mishra[3]

1 School of Medicine and Public Health, University of Newcastle, Newcastle, New South Wales, Australia, 2 Hunter New England Local Health District, John Hunter Hospital, New Lambton, New South Wales, Australia, 3 School of Public Health, The University of Queensland, Herston, Queensland, Australia

* julie.byles@newcastle.edu.au

## Abstract

Health assessments have potential to improve health of older people. This study compares long-term health care utilisation, physical functioning, and mortality for women aged 75 years or over who have had a health assessment and those who have not. Prospective data on health service use, physical functioning, and deaths among a large cohort of women born 1921–26 were analysed. Propensity score matching was used to produce comparable groups of women according to whether they had a health assessment or not. The study population included 6128 (67.3%) women who had an assessment, and 2971 (32.7%) women who had no assessment. Propensity matching produced 2101 pairs. Women who had an assessment had more use of other health services, longer survival, and were more likely to survive with high physical functioning compared to women with no assessment. Among women who had good baseline physcial functioning scores, women who had an assessment had significantly lower odds of poor outcomes at 1000 days follow-up compared to women who had no assessment (OR: 0.67, 95%CI: 0.52, 0.85). This large observational study shows the real-world potential for assessments to improve health outcomes for older women. However, they also increased health service use. This increased healthcare is likely to be an important mechanism in improving the women's health outcomes.

## Introduction

Age is commonly associated with decline in physical functioning, with many older people experiencing decreased ability to maintain independence in their activities of daily living and instrumental activities. Comprehensive health assessment with appropriate follow-up of identified needs is one option for prevention of functional decline and other adverse events among older people, and improving their quality of life [1–3]. The basis for these assessments is that older people experience many preventable problems that would otherwise go undetected, and

**Data Availability Statement:** Please note use of linked administrative and survey data for ALSWH projects is strictly regulated. This strict restriction is due to the inclusion of data from the Australian Government Medicare database, with the ethics and legal approvals requiring that the data can only be accessed by approved researchers and only on site within the ALSWH offices. There are therefore legal and ethical restrictions on sharing the de-identified data set. De-identified survey data are available to collaborating researchers where a formal request to make use of the material has been approved by the ALSWH Data Access Committee. The committee is receptive of requests for datasets required to replicate results. Information on applying for ALSWH data is available from https://www.alswh.org.au/for-data-users/applying-for-data/. In order for linked data provided by third parties to be accessed through ALSWH, every data user must be added to the applicable Data Use Agreements and Human Research Ethics Committee protocols. Data Access enquiries can also be directed to this email: alswh@uq.edu.au.

**Funding:** The Australian Longitudinal Study on Women's Health (ALSWH) is funded by the Australian Government Department of Health – Grant Number: G2000719 awarded to JEB and GM. JEB, MAT, CLC, TM, BRN and GM received funding for this research from the National Health and Medical Research Council (NHMRC) – Grant Number: G1400038. The funders had no role in study design, data collection and analysis, decision to publish, or preparation of the manuscript.

**Competing interests:** The authors have declared that no competing interests exist

that small improvements in lifestyle, support and clinical care may result in major functional or quality of life gains across the course of later life [2,3]. The evidence-base for the effectiveness of such health assessments is largely from studies in Europe and the United States of America, with mixed evidence of effectiveness. However, the majority of trials demonstrate improvement in health status for those who have assessments [1,3–5].

In their systematic review of 18 trials, Stuck et al. [3] found that multidimensional assessment with follow-up resulted in less functional decline. There was also a significant protective effect on mortality for people aged in their 70s but a non-significant effect for those in their 80s. Likewise, assessments were more effective among the subset of the trial populations who had lower mortality rates than among those with higher mortality risk. Similar findings were identified by Beswick et al. [1] who reviewed 28 trials of geriatric assessment for people in the community, and 24 trials where assessments were targeted to frail older people. The pooled effect of these trials showed better physical function among people who had assessments, particularly when assessments were generally targeted to people in the community. However, in this review, there was no clear evidence that assessments reduced mortality rates [1].

Another systematic review [6], examined comprehensive geriatric assessments as one component of non-pharmacological interventions for community dwelling older people. The authors identified 15 studies which included assessment, across a pooled sample of 4603 people.

The assessments evaluated in these studies generally involved medical, psychosocial or functional assessments to inform care plans for primary care and rehabilitation. Only three studies demonstrated statistically significant effects of the health assessments [7–9] and overall, the effects were small or negligible. Most of these studies were aimed at apparently well older people in the community with low rates of disability, and the authors consider the possibility that there was little opportunity to detect differences in disability rates across the study periods [6].

The overall consensus from these studies is that health assessments should have potential to improve outcomes for older people, however there are very mixed results across multiple studies and reviews involving different populations and approaches to the assessments and their follow-up. In this study, we used observational data to examine the effects of assessments across a population of older adults, within the context of the Australian health care system. The Australian government has been subsidising annual assessments for people aged 75 years or over since 1999. These assessments are designed to enhance the capacity of primary health care to improve the health and quality of life of older people with complex care needs. They provide an opportunity for in-depth assessment of the person's health and allow preventive health care and education. Specific aspects of the assessment include patient's blood pressure; pulse rate and rhythm; medications; continence; immunisation status for influenza, tetanus and pneumococcus; physical function, including the patient's activities of daily living, and whether or not the patient has had a fall in the last three months; psychological function, including cognition and mood; oral health; nutrition; and social function, including the availability and adequacy of paid and unpaid help, and whether the person is caring for another person.

The assessments could potentially also increase case finding and older person's access to health services, enabling greater collaboration between general practitioners (GPs), nurses and other health professionals [10,11].

A randomised controlled trial showed that health assessments for older people in Australia were associated with small improvements in quality of life outcomes, but there was no significant difference in the probability of hospital admission or death [12].

Since their introduction in 1999, the health assessments have been widely used by older people across Australia [13–15], with evidence of their impact on the identification and

management of health problems [16–18]. However, our previous research into the uptake of health assessments by older Australian women shows that those who have health assessments are already having more GP visits, and also that few women have repeat assessments as intended by the program [14].

The objective of the current study was to undertake a long-term evaluation of the effect of the health assessments on survival and physical functioning, by comparing outcomes for older women who have had at least one health assessment with those who have never had an assessment, using propensity score matching. We hypothesised that having at least one health assessment after the age of 75 leads to better survival and improved physical functioning. We also expected that health assessments would be associated with higher levels of health care use, either as a marker of better quality care, and/or as the mechanism by which needs identified during the assessment process might be addressed. Based on earlier reviews, and on the premise that earlier intervention would have larger effects, we also expected that assessments would be more effective in improving survival and physical functioning for women who began with better physical functioning at baseline compared to women who were already in poor health, and for those with lower mortality risk.

## Materials and methods

### Data source

We used data from the Australian Longitudinal Study on Women's Health (ALSWH), which is a prospective study of a large cohort of older women. The women were recruited in 1996 when they were aged 70–75 years (N = 12432) and have completed self-report surveys of health and wellbeing on a three-yearly basis up until survey 6 in 2011 and six-monthly thereafter [19]. The women's survey data are routinely linked to data from the Medicare Benefits Scheme (MBS) which is Australia's universal health insurance scheme. These data provide information on dates of visits to GPs (unreferred visits), medical specialists (referred visits), tests, and health assessments. ALSWH participants were eligible to be included in this study if they had not opted out of data linkage, had at least one unreferred doctor's visit recorded in MBS between 1 November 1999 and 31 December 2013, and completed Survey 2 in 1999.

### Outcome variables

The primary outcomes were mortality rates and poor physical functioning. Date of death was provided by the National Death Index (NDI) [20] for deaths up to November 2013. Physical functioning was assessed from the SF-36 physical functioning (PF) subscale included in each survey.

The PF subscale scores range from 0–100 with lower values representing poorer functioning [21]. For analysis in this study, scores were categorised as <42 poor, 42–68 fair, >68 good, based on previous analysis of data from these women [22]. Outcomes of death or poor physical functioning on the most recent survey were combined to provide a single indicator of poor health. Health care utilisation was assessed as a secondary outcome, and was defined as the number of GP visits, specialist visits and pathology/diagnostic testing per six-month period as identified in the MBS data set.

### Explanatory variable and covariates

The main explanatory variable was having a health assessment (MBS items: 700–710, 712–719) based on claims in the MBS data. Covariates were baseline age, area of residence (metropolitan or non-metropolitan), education, marital status, difficulty managing on income, private health

insurance, baseline SF-36 physical functioning score, SF-36 mental health index, smoking, Body Mass Index, multimorbidity, symptoms, and number of visits to a general practitioner. Multimorbidity was determined by summing the number of self-reported diagnoses of diabetes, heart disease, asthma, arthritis, hypertension or stroke, and categorised as 0–2 and >2. Symptoms included: stiff or painful joints; breathing difficulty; chest pain; any of leaking urine, needing to rush to the toilet to pass urine, passing urine more than twice during the night; dizziness or loss of balance; difficulty seeing a newspaper, even with glasses; wear a hearing aid; and any self-report of having slipped, tripped, stumbled, had a fall to the ground, or been injured as a result of a fall in the last 12 months.

## Propensity score matching

For matching, participants who had an assessment were assigned into blocks based on the date of their first health assessment and corresponding to the interval between surveys (Fig 1). Propensity scores [23] were generated using logistic regression models and used to match participants who had never had an assessment with participants who had an assessment if they had a similar propensity score (within a 0.02 calliper distance matching algorithm), and if they had at least one unreferred visit in the same block. Matched pairs were retained only if the "no assessment" match had not died within three months of the health assessment date. Women who had an assessment but who could not be matched were excluded from the analysis.

Unmatched women from the no assessment group were placed in the remaining subset for matching with women who had an assessment in the next block. This process was repeated for all five blocks. Covariate values for propensity scores were from the survey representing the

| **Time period** | | | | |
|---|---|---|---|---|
| November 1999-February 2002 (Survey 2) | March 2002 – February 2005 (Survey 3) | March 2005 - February 2008 (Survey 4) | March 2008-February 2011 (Survey 5) | March 2011 - December 2013 (Survey 6) |
| **Block 1**: 1102 women had assessment; 1102 never had an assessment | | | | |
| | **Block 2**: 205 women had assessment; 205 never had an assessment | | | |
| | | **Block 3**: 205 had assessment; 205 never had an assessment | | |
| | | | **Block 4**: 78 assessment; 78 never assessment | |
| | | | | **Block 5**: 23 assessment; 23 never |

**Fig 1. Blocks and matched pairs for comparisons.**

start of the block. Models included 24 covariates (see Table 1), two squared terms, and between 8 to 66 interaction terms across the cohort blocks (See S1 Table). The c-statistic was used as a measure of propensity score performance through the model building process [24].

## Statistical analysis

Characteristics were compared for differences across groups for each continuous and categorical variable (t-tests and $\chi^2$-tests, respectively). Survival time was calculated as the number of follow-up days from the start of the cohort block to either the date of death, or the last follow-up date (December 2013). We used both parametric and semi-parametric regression models of survival estimates across time and assessment status. Our full survival analysis used Multivariable Cox Survival models to deal with the long-term follow-up and non-proportional hazards, and with robust variance estimation to account for stratified blocks. Forward selection of each of the covariates was based on univariable analysis (Step 1) and then the interaction with time since the health assessment was added (Step 2).

A covariate by time interaction was considered significant if the likelihood ratio test for the model including this interaction term indicated a significantly better fit compared to the model without this interaction. In the last step (Step 3) we assessed the time-dependent effect of each covariate, its interaction with having a health assessment (or not), and the interaction between the covariate and assessment and time.

The final Cox model was then used to compute a prognostic score for each participant using the methods of Putter, Sasako, Hartgrink, van de Velde and van Houwelingen [25]. The score consisted of the sum of all baseline variables (excluding time-dependent variables) which significantly predicted the hazards of death, multiplied by the estimated coefficients for these baseline variables. Participants were then grouped in three equal groups based on the prognostic score corresponding to level of risk (low, medium, high). Overall survival curves were then plotted depending on the risk. Kaplan-Meier survival curves were estimated separately for women who had and had not had an assessment, and by physical functioning categories.

Logistic regression models with forward elimination techniques were used to assess the association between having an assessment and the composite outcome [poor physical function/death]. The Hosmer and Lemeshow goodness-of-fit test was used to assess the fit for each model, and the Wald test using odds ratios (95% Confidence Interval) was used to consider variable fit. Akaike Information Criteria were used to compare between models. All analyses were performed using SAS v9.4 (x64), SAS Institute, Inc., Cary, NC.

## Ethics approvals

This project has ongoing ethical clearance from the University of Newcastle (H-076-0795) and the University of Queensland (2004000224) Human Research Ethics Committees.

Ethical approval for the linkage of ALSWH survey data to the Medicare Benefits Scheme and to the National Death Index was received from the Australian Institute of Health and Welfare Research Ethics Committee (EC 2012/1/12) and registered with the University of Newcastle.

## Results

A total of 9,099 women met all eligibility criteria and were classified as having had a health assessment (N = 6128, 67.3%) or never having a health assessment (N = 2,971, 32.7%). A total of 4,202 participants were matched using propensity score analysis.

Table 1 compares Survey 2 characteristics for women having assessments and no assessments for the whole sample and for the matched pairs. Prior to propensity score matching

**Table 1. Characteristics of women who have and have not had assessments, before and after matching.**

| | Characteristics at Survey 2 | | | Characteristics after matching | | |
| --- | --- | --- | --- | --- | --- | --- |
| | N = 9,099 women | | | N = 2,101 pairs (4202 women) | | |
| | No Health Assessment | Health Assessment | P-Value | No Health Assessment | Health Assessment | P-Value |
| | N = 2971 32.7% | N = 6128 67.3% | | N = 2,101 50% | N = 2,101 50% | |
| **Marital Status** | | | | | | |
| Not Widowed | 55.8% | 64.6% | | 60.2% | 61.6% | |
| Widowed | 44.2% | 35.4% | <0.001 | 39.8% | 38.4% | 0.975 |
| **Smoking Status** | | | | | | |
| Never smoked | 60.8% | 63.7% | | 61.8% | 61.9% | |
| Ex/Current | 39.3% | 36.3% | 0.009 | 38.2% | 38.1% | 0.799 |
| **Alcohol consumption**\* | | | | | | |
| Non-drinker | | | 0.529 | | | 0.932 |
| Low/rarely drinker | 34.2% | 33.1% | | 31.8% | 31.4% | |
| Risky/high risk drinker | 62.2% | 63.4% | | 64.5% | 64.8% | |
| **Country of Birth**\* | | | | | | |
| Australian born | | | 0.006 | | | 0.151 |
| Other English speaking background | 79.6% | 76.5% | | 79.7% | 76.7% | |
| European | 11.6% | 14.2% | | 12.5% | 14.9% | |
| Other | 7.1% | 7.6% | | 7.1% | 7.2% | |
| **Educational Qualifications** | | | | | | |
| Up to Year 10 | 71.1% | 71.4% | | 68.2% | 68.2% | |
| Post-school | 28.9% | 28.6% | 0.154 | 31.8% | 31.8% | 1 |
| **Region** | | | | | | |
| Major City | 67.5% | 65.5% | | 60.6% | 60.6% | |
| Regional/remote | 32.6% | 34.6% | 0.057 | 39.4% | 39.4% | 0.801 |
| **Private Health insurance** | 46.9% | 51.8% | <0.001 | 47.0% | 45.9% | 0.665 |
| **Manage on Income** | | | | | | |
| Not too bad/Easy | 76.4% | 73.7% | 0.007 | 76.1% | 75.6% | 0.916 |
| **General Health** | | | | | | |
| Good/excellent | 68.4% | 74.4% | | 72.7% | 72.0% | |
| Fair/poor | 31.6% | 25.6% | <0.001 | 27.3% | 28.0% | 0.947 |
| **Visits to the GP** | | | | | | |
| Up to four/year | 40.7% | 39.7% | | 46.6% | 44.4% | |
| More than four | 59.3% | 60.4% | 0.35 | 53.4% | 55.6% | 0.732 |
| **Body Mass Index** | | | | | | |
| Normal | 52.3% | 49.4% | | 53.3% | 52.5% | |
| Underweight | 4.3% | 2.9% | | 2.2% | 2.6% | |
| Overweight/Obese | 43.5% | 47.6% | <0.001 | 44.5% | 44.9% | 0.975 |
| **Co-morbidities** | | | | | | |
| Diabetes | 7.9% | 7.0% | 0.132 | 6.6% | 6.5% | 0.413 |
| Heart disease | 14.8% | 12.4% | 0.002 | 13.6% | 13.6% | 0.352 |
| Asthma | 13.3% | 12.7% | 0.456 | 12.5% | 12.7% | 0.891 |
| Arthritis | 40.6% | 42.7% | 0.059 | 41.1% | 43.1% | 0.925 |
| Hypertension | 33.1% | 34.7% | 0.13 | 34.4% | 35.9% | 0.592 |
| Stroke | 4.1% | 2.4% | <0.001 | 3.3% | 3.3% | 1 |
| **Symptoms** | | | | | | |
| Pain in the joints | 43.8% | 46.3% | 0.029 | 43.8% | 45.0% | 0.877 |
| Heart Symptoms | 2.3% | 2.3% | 0.908 | 1.7% | 2.9% | 0.317 |

*(Continued)*

**Table 1.** (Continued)

| | Characteristics at Survey 2 | | | Characteristics after matching | | |
|---|---|---|---|---|---|---|
| | N = 9,099 women | | | N = 2,101 pairs (4202 women) | | |
| | No Health Assessment | Health Assessment | P-Value | No Health Assessment | Health Assessment | P-Value |
| | N = 2971 32.7% | N = 6128 67.3% | | N = 2,101 50% | N = 2,101 50% | |
| Urine Leakage | 2.1% | 1.5% | 0.054 | 1.5% | 1.9% | 0.292 |
| Feeling dizzy | 3.8% | 3.5% | 0.373 | 3.5% | 3.8% | 0.621 |
| Hearing Problems | 13.6% | 14.5% | 0.26 | 11.0% | 11.4% | 0.962 |
| Vision Problems | 27.6% | 25.0% | 0.009 | 22.6% | 22.9% | 0.855 |
| Falls | 40.0% | 36.7% | 0.801 | 38.4% | 39.2% | 0.752 |
| **Need help with daily tasks**[*] | 9.6% | 5.4% | <0.001 | 7.2% | 6.5% | 0.398 |
| | Mean | Mean | | Mean | Mean | |
| | (Median) | (Median) | | (Median) | (Median) | |
| **Age** | 75.4 (75.4) | 75.3 (75.2) | 0.007 | 76.53 (2.8) | 76.5 (2.8) | 0.976 |
| **Mental health Score** | 78.4 (84.0) | 79.6 (84.0) | 0.015 | 78.5 (16.0) | 78.6 (16.0) | 0.902 |
| **Physical Functioning Score** | 60.3 (65.0) | 64.4 (70.0) | <0.001 | 56.6 (26.6) | 56.6 (26.6) | 0.996 |

[*] alcohol consumption, country of birth and need help with daily tasks were not included in propensity scores.

there were significant differences between participants who did and did not have an assessment. After propensity score matching, no statistically significant differences were evident between participants who did and did not have an assessment. The c-statistic values for the propensity models range around 0.5 indicating that each variable in the propensity score models is reasonably predictive of having an assessment. Variables country of birth, alcohol and needing help with daily living were not included in propensity models but were also balanced across the matched groups.

Women who had an assessment were less likely to have died over the follow-up period. Among all women, 62.7% of those who did not have a health assessment died, and 44.3% of those who did have an assessment died. Among the matched pairs, 66.2% of those who did not have a health assessment died, and 59.9% of those who did have an assessment died. However, 120 of the matched women died within three months of the health assessment date for the pair, and these 120 pairs were excluded. This left 3,204 women in block 1 (S2-23), 410 in block 2 (S3-S4), 156 in block 3 (S4-S5), 146 in block 4 (S5-S6) and 46 in block 5 (S6-S7) (Fig 1). The mean follow-up across all blocks was 8.5 years.

For Cox models assessing time to death, the proportional hazards assumption was violated for the health assessment variable, implying that the hazards varied with time since having the first health assessment. All covariates were found to be significant on univariable analysis and were entered into the multivariable Cox regression model using a stepwise process (Step 1). However, in the multivariable models, and once interactions with time (Step 2), and the interaction between *assessment* and *time* were added (Step 3), a number of covariates and interactions did not improve the model fit and these covariates were removed from further modelling.

Table 2 shows parameter estimates for Cox models for univariable analysis (Step 1), adjusting for significant interactions between covariates and time (Step 2), and finally the effects of assessments after all other significant variables have been included in the model.

In these analyses, the hazards of death increased with age, more GP visits at baseline, multi-morbidity, and poor physical functioning, and decreased for those having an assessment. The estimated hazard ratio associated with having a health assessment as a function of time $t$ was

**Table 2. Predictors of time to death, based on stratified proportional hazards analysis.**

| Variable | Step 1 | P-Value | Step 2: Treatment Effect | P-Value |
|---|---|---|---|---|
| | Parameter Estimate (Standard Error) | | Parameter Estimate (Standard Error) | |
| **Age** | | | | |
| 75–80 years | Reference | | Reference | |
| >80–85 years | 0.266 (0.051) | < .0001 | 0.266 (0.051) | < .0001 |
| >85 years | 0.448 (0.153) | | 0.411 (0.153) | 0.007 |
| **GP** | | | | |
| Less than four | Reference | | Reference | |
| More than four | 0.143 (0.049) | 0.004 | 0.147 (0.049) | 0.003 |
| **Area of residence** | | | | |
| Non-urban living | Reference | | Reference | |
| Urban living | -0.117 (0.046) | 0.011 | -0.118 (0.046) | 0.01 |
| **CoMorbidities** | 0.086 (0.023) | <0.001 | 0.094 (0.024) | <0.001 |
| **Problems with Sight** | | | | |
| No | Reference | | Reference | |
| Yes | 0.072 (0.055) | 0.19 | 0.083 (0.054) | 0.127 |
| **Feeling Dizzy** | | | | |
| No | Reference | | | |
| Yes | 0.08 (0.099) | 0.419 | - | - |
| **Urinary Incontinence** | | | | |
| No | Reference | | | |
| Yes | 0.122 (0.144) | 0.394 | - | - |
| **Joint Pain and Symptoms** | | | | |
| No | Reference | | Reference | |
| Yes | -0.102 (0.049) | 0.038 | -0.097 (0.049) | 0.046 |
| **Physical function** | | | | |
| Not Fair functioning | Reference | | Reference | |
| Fair functioning | 0.348 (0.513) | 0.498 | 0.331 (0.513) | 0.518 |
| Not Poor functioning | Reference | | Reference | |
| Poor functioning | 1.674 (0.471) | <0.001 | 1.753 (0.469) | <0.002 |
| **Fair physical functioning by time** | | | | |
| Fair functioning*log(t+1) | -0.036 (0.112) | 0.749 | -0.032 (0.111) | 0.771 |
| **Poor physical functioning by time** | | | | |
| Poor functioning*log(t+1) | -0.229 (0.103) | 0.027 | -0.244 (0.103) | 0.018 |
| **Treatment (health assessment)** | - | - | -3.508 (0.442) | <0.001 |
| **Treatment by time** | - | - | 0.693 (0.096) | <0.001 |
| **AIC** | 29857.018 | | 29731.752 | |

given by: HR(t) = exp(-3.51 + 0.69t). Hazard ratios were 0.06, 0.24, and 0.99 at one, three and five years, respectively suggesting that assessment has an initial protective effect once time-by-treatment interactions was included. However, the parameter associated with having a health assessment and time interaction was positive, suggesting that the hazard ratios were increasing over time. This indicates that the protective effect of having a health assessment diminishes over time.

To further understand which women might have the most potential to benefit from assessments, and to account for heterogeneity in survival probabilities, we calculated prognostic scores based on the significant non-time dependent variables using the methods described by

Putter et al. [25].

$$y = Age(>75 - 80years) * 0.2663 + Age(>80 + years) * 0.4113 + Urban * (-0.1176) \\ + Morbidities * 0.0938 + GP\ visits * 0.1472 + Joints * (-0.0973)$$

These prognostic scores were then used to divide the women into equal sized groups corresponding to low, medium and high risk of death. Women in the low risk group had a median survival time from baseline (Survey 2, 1999) of 178.3 months [IQR:116.9, 178.3], medium risk 147.5 months [IQR: 94.1, 178.3] and women of high risk had the shortest median survival time of 111.8 months [IQR:70.1, 163.4]. The mean age at baseline was 74.8 (SD:1.51) in the low risk group, 76.4 (SD: 2.07) in the medium risk group, and 78.4 (SD: 3.24) in the high risk group (overall mean 76.0, Std Dev: 2.4–3.5 years).

Women in the high risk group had a mean baseline physical functioning score (51.9, SD: 26.6) while women in the low and medium risk group had substantially higher physical functioning scores (67.9, SD: 23.8 and 61.3, SD: 25.8, respectively). Accordingly, there was a greater proportion of women in the high-risk group with poor physical functioning (37.2%) compared to the medium (24.3%) and low (16.8%) risk groups. Women from the high risk group also had a mean of 1.8 (95%CI:1.7–1.9) morbidities compared to women in the low and medium group (0.6 CI:0.6–0.6 vs 1.2 CI:1.1–1.2, respectively).

Among the medium and high-risk groups, a greater proportion of women lived in non-urban areas compared to women in the low risk groups (59.8%, 72.1% and 47.5%, respectively). Women in the high-risk group were more likely to have had more than four GP visits in the last 12 months. Given the women were matched, there were no differences in the proportions of women receiving health assessments across the three risk groups. However, women in the low risk group were more likely to be younger at the time of first assessment compared to women in the medium risk and high-risk groups. Approximately 343 (25.1%) women in the high-risk group were 80 years or older when they had their first health assessment (S2 Table).

Fig 2 shows the survival curves for women in the low mortality risk groups, their baseline physical functioning category (good, fair and poor), and whether they had an assessment or not. Approximately 70% of women grouped as low risk with good physical functioning who had a health assessment survived till the end of the study, compared to 59% of women with the same risk profile who did not have an assessment. Women in the low risk group who had fair or poor physical functioning had survival probabilities of 67% and 50%, respectively if they had an assessment and 55% and 34% respectively, if they did not have an assessment. Similar difference in survival probabilities were seen in the medium risk group (S1A Fig), although their overall survival time was less.

Among women in the high-risk groups, marginal differences were seen between assessment and no assessment for women having good or fair physical functioning (S1B Fig).

Results of the multivariable logistic regression models of the association between health assessments and the composite outcome (poor physical functioning or death) at 1000 days and 2000 days follow-up are shown in Table 3.

There was a signficant interaction between physical functioning assessed at the start of the relevant block, having a health assessment, and poor outcome. Participants who had good baseline physcial functioning scores and who had an assessment had significantly lower odds of poor outcomes at 1000 days follow-up compared to participants having an unreferred visit only (OR: 0.67, 95%CI: 0.52, 0.85). There were no differences between assessment and no assessment groups for participants having fair or poor physcial functioning at baseline. Other factors that were protective against a poor outcome included living in a major metropolitan area (OR: 0.83 95%CI: 0.71, 0.97) and having excellent, very good or good self-rated health (OR: 0.529, 95%CI: 0.442, 0.634). Each one year increase in age resulted in a 6% increase in odds of poorer outcomes at 1000 days follow-up.

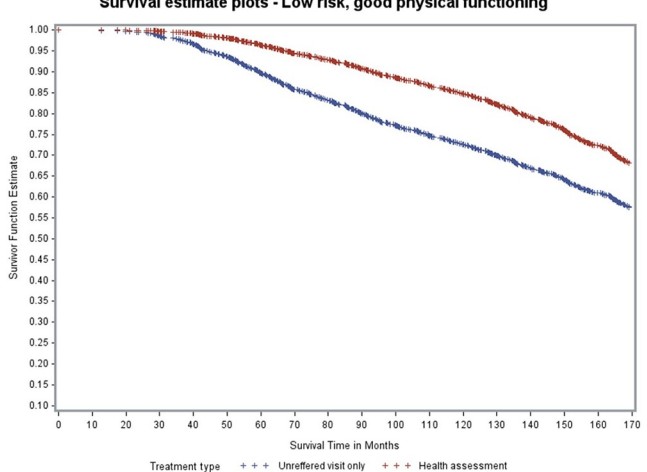

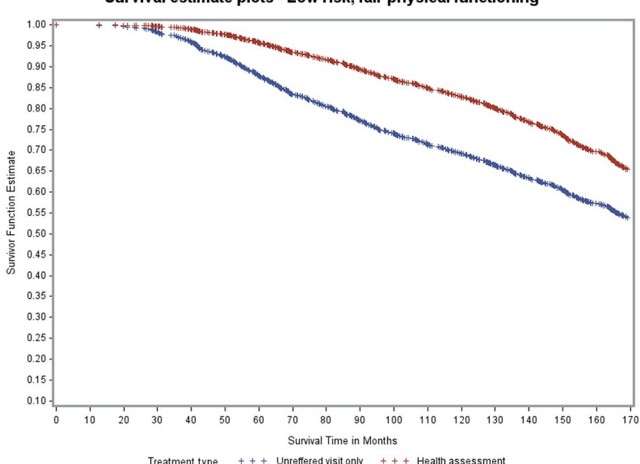

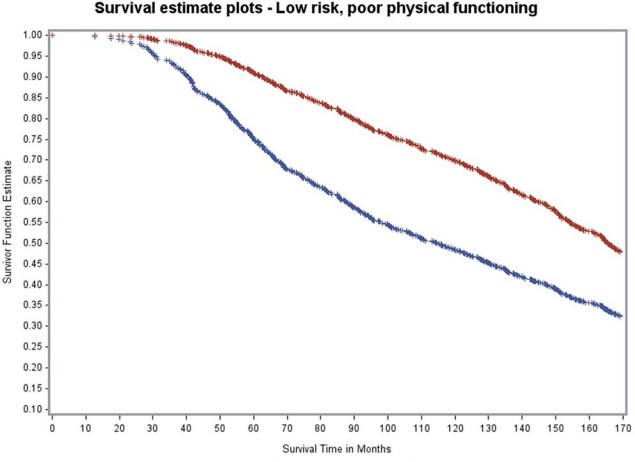

**Fig 2. Model-based survival curves for women with low mortality risk.**

At 2000 days from assessent, participants with fair functioning at baseline who had a health assessment had almost 40% reduced odds of poor health outcome compared to women who only had an unreferred visit (OR:0.67, 95%CI:0.51, 0.87). Similar results were seen for those with poor physical (OR:0.49, 95%CI: 0.32, 0.77). Higher odds of poor outcomes at 2000 days were also observed for women who were overweight (OR: 1.59, 95%CI: 1.01, 2.50), and those with two or more conditions (OR: 1.17, 95%CI:1.01, 1.37).

The mean difference in health service utilisation between participants who had an assessment and those who had an unreferred visit only is shown in Fig 3 (Blocks 1, 2 and 3) and S2 Fig (Blocks 4 and 5). Regardless of the cohort block, participants who had an assessment had significantly higher use of health services both before and after the assessments, and a further increase in the three years following the assessments.

## Discussion

This large observational study used propensity score matching to compare groups of women who had and had not had assessments in terms of survival, physical functioning, and health service use.

**Table 3. Predictors of poor physical functioning/death by 1000 and 2000 days.**

| Description | At 1000 days | At 2000 days |
|---|---|---|
| Physical functioning at the start of the block | OR [95%CI] | OR [95%CI] |
| **Good functioning** | | |
| Unreferred visit only | Reference | Reference |
| Health assessment | 0.665, [0.519, 0.853] | 0.964 [0.791, 1.175] |
| **Fair functioning** | | |
| Unreferred visit only | Reference | Reference |
| Health assessment | 0.991, [0.776, 1.264] | 0.665 [0.512, 0.865] |
| **Poor functioning** | | |
| Unreferred visit only | Reference | Reference |
| Health assessment | 0.710, [0.501, 1.007] | 0.494 [0.317, 0.77] |
| **Perceived General Health** | | |
| Fair/poor | Reference | Reference |
| Excellent/good/very good | 0.529, [0.442, 0.634] | 0.556 [0.455, 0.681] |
| **Area of Residence** | | |
| Non-Urban | Reference | Reference |
| Urban | 0.828, [0.706, 0.971] | 0.932 [0.801, 1.085] |
| **Age in years** | 1.064, [1.008, 1.124] | 1.13 [1.073, 1.19] |

At 1000 days: Adjusted for Urinary Incontinence.

At 2000 days: Adjusted for BMI, number of comorbidities, problems with sight, problems with hearing, falls in the past 12 months.

Women who had health assessments had better survival and were less likely to have the composite outcome of poor health or death. These findings add to the evidence from numerous large randomised controlled trials to show that health assessments can improve health outcomes. Beyond these previous trials, the study also shows how assessments are used at a

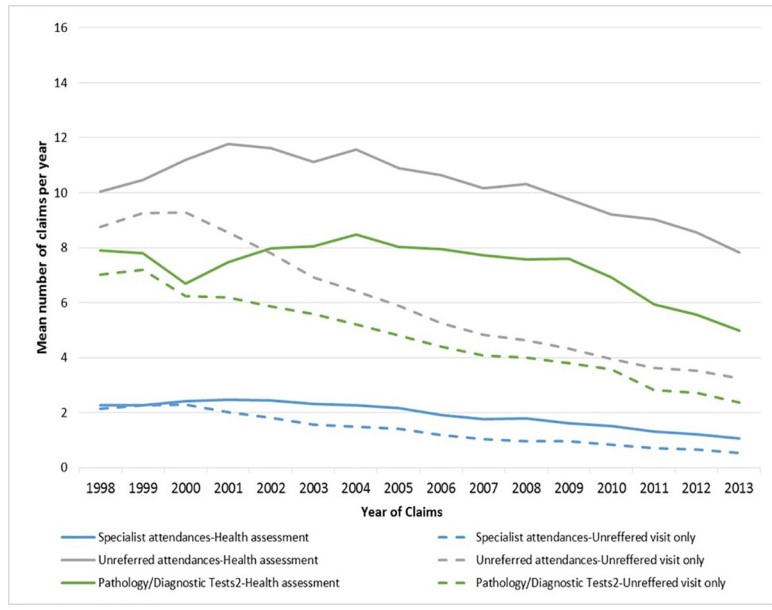

**Fig 3. Health service utilisation for women who had an assessment and those who did not (Blocks 1,2,3).**

population level and over time, as well as their potential effects on health outcomes. Unlike earlier studies where assessments were conducted within the context of a trial with protocols for the administration of the assessment and follow-up [12], this study evaluates assessments according to their uptake by women and to how they are delivered by general practitioners within the health system. While this observational approach provides evidence as to how assessments perform in practice, there are limitations in terms of the study rigour compared to an experimental trial design including the potential for incomplete matching of assessed and non-assessed groups (discussed later). The external validity of both trial and observational approaches can be limited, with the former affected by participation bias, and the latter potentially limited by matching, as seen in this study. In this study the matched pairs were on average one year older than the overall group of women who had assessments, they also had fewer GP visits and lower physical functioning scores.

The differences in survival may be explained by the higher overall health care use by the women who had assessments, both before and after the assessment. It may be that women who have assessments are those who are in more frequent contact with the health care system [14], and receive better quality care. We did observe an increase in health care use following the assessments suggesting also that more intensive health care use may be a mechanism for improved health outcomes among assessed women. This is consistent with a previous review that concluded that health assessment programs were particularly effective if they incorporated ongoing follow-up [3]. In their review, Beswick et al. [1] discuss the discrepancy between their findings and the results of a large MRC trial which found that assessments were not effective. The authors of the MRC trial note that assessments were not generally well conducted, and this may have limited their effectiveness [26].

Beswick et al. [1] also consider the effectiveness of the health system overall as an explanation for this discrepancy between the results of controlled studies and the MRC trial.

More recent research has shown some additional evidence for the potential for health assessments to reduce mortality rates. A randomised controlled trial in Germany showed a 20% reduction in mortality rates for those who had a health assessment [27]. Also, a recent retrospective analysis using linked MBS and aged care data to assess the use and impacts of health assessments for people using aged care, showed that after propensity score matching, people who had an assessment had 7% lower risk of death with a mean follow-up of 1.9 years. However, those who received assessments had a higher risk of admission to permanent residential aged care [28].

In our study, the logistic regression analysis showed that those with good physical functioning were less likely to have poor outcomes at 1000 days if they had an assessment. This is consistent with the review by Beswick et al. [1], which found more evidence for the effectiveness of assessments for older people in the general community than for assessments targeted to frail older people. However, few trials follow people for as long as 2000 days. At 2000 days, we found that there were significantly lower odds of poor health outcome for those with fair or poor physical functioning if they had an assessment. Assessments may have a particular role for improving outcomes for those with poor physical functioning but long survival probabilities. This possibility is supported by our analysis showing that where women had low or medium mortality risk, there were larger survival differences between women who had an assessment and those who did not, even if they had poor physical functioning.

That assessments on their own would have such long lasting effects in frail older people is unlikely, but it may be that they set people on a better trajectory of health service use.

It should however also be noted that people with higher physical functioning scores had better chances of survival and these survival probabilities were higher if people had an assessment.

In undertaking these analyses we have been extremely careful to avoid immortality bias, whereby the women have to survive long enough to have assessments (and therefore have an

artefactual survival advantage), using propensity matching, removing matches who died within three months of the assessment date, and using prognostic stratification.

Propensity matching also increases the comparability of the groups in terms of other factors which may affect survival or quality of life. Propensity methods are increasingly used in clinical research to enable comparisons of treated and untreated populations using observational data [29]. Our approach using 1:1 matching, has an advantage of producing balanced groups for known confounders, however there is a risk that some residual confounding may still bias the results, and propensity matching cannot account for unknown confounders in the same way that a randomised trial can. A disadvantage of 1:1 matching is that the resultant matched pairs may differ from the original population.

However, for women having health assessments, the differences between matched an unmatched samples in our study are small, due to the large pools available for matching. The matched sample were slightly older and better educated, had poorer median physical functioning score, had fewer GP visits, and included fewer non-smokers, more people in non-metropolitan areas, and fewer people with private health insurance.

Other strengths of this study include the long-term follow-up and complete outcomes for a large cohort of women in terms of health service use and death.

Ascertainment of physical functioning outcomes was less complete as this was dependent on participation in the survey. Attrition rates for ALSWH are around 10% for each survey and have been reported elsewhere [30–33].

## Conclusion

Health assessments have potential to improve survival and physical functioning of older people, but their uptake and their effect appears to be strongly coincident with greater use of health services. If assessments are effective in improving health care use and outcomes, then attempts to extend assessments to reach underserved women may increase their overall effectiveness.

The value of assessments may be strongly linked to the quality and effectiveness of health services overall. Assessments may be a key gateway to better care, and it is important to understand the key elements of effective care for older people and how they may be improved through the assessment process.

## Supporting information

**S1 Fig.** a. Model-based survival curves for women with medium mortality risk. b. Model-based survival curves for women with high mortality risk.
(PDF)

**S2 Fig. Health service utilisation for women who had an assessment and those who did not (Blocks 4,5).**
(PDF)

**S1 Table. Interaction terms for propensity matching.**
(PDF)

**S2 Table. Characteristics of women among the low, medium and high-risk groups.**
(PDF)

## Acknowledgments

The research on which this paper is based was conducted as part of the Australian Longitudinal Study on Women's Health by the University of Queensland and the University of

Newcastle. We are grateful to the women who provided the survey data. The authors acknowledge the Department of Health and Medicare Australia for providing MBS data, and the Australian Institute of Health and Welfare (AIHW) as the integrating authority. We also acknowledge AIHW and State and Territory Death Registries for the National Death Index.

## Author Contributions

**Conceptualization:** Julie E. Byles.

**Formal analysis:** Xenia Dolja-Gore.

**Methodology:** Xenia Dolja-Gore, Julie E. Byles.

**Supervision:** Julie E. Byles, Balakrishnan R. Nair, Gita D. Mishra.

**Writing – original draft:** Xenia Dolja-Gore, Julie E. Byles.

**Writing – review & editing:** Xenia Dolja-Gore, Julie E. Byles, Meredith A. Tavener, Catherine L. Chojenta, Tazeen Majeed, Balakrishnan R. Nair, Gita D. Mishra.

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
