## [Decision Letter · Decision Letter 0]

14 Dec 2020

PONE-D-20-34902

Estimating the effect of health assessments on mortality, physical functioning and health care utilisation for women aged 75 years and older

PLOS ONE

Dear Dr. Byles,

Thank you for submitting your manuscript to PLOS ONE. After careful consideration, we feel that it has merit but does not fully meet PLOS ONE’s publication criteria as it currently stands. Therefore, we invite you to submit a revised version of the manuscript that addresses the points raised during the review process.

We look forward to receiving your revised manuscript.

Kind regards,

Yiqiang Zhan

Academic Editor

PLOS ONE

Journal Requirements:

'..We are grateful to the Australian Government Department of Health for funding..'

'The author(s) received no specific funding for this work'

Reviewers' comments:

Reviewer's Responses to Questions

**Comments to the Author**

1. Is the manuscript technically sound, and do the data support the conclusions?

Reviewer #1: Partly

Reviewer #2: Yes

2. Has the statistical analysis been performed appropriately and rigorously? 

Reviewer #1: Yes

Reviewer #2: Yes

3. Have the authors made all data underlying the findings in their manuscript fully available?

Reviewer #1: Yes

Reviewer #2: No

4. Is the manuscript presented in an intelligible fashion and written in standard English?

Reviewer #1: Yes

Reviewer #2: Yes

5. Review Comments to the Author

Reviewer #1: #0. What is the strengths and the limitations of this study comparing previous studies to found that health assessment improved the health of older people?

Please clarify it.

(Strengths and limitations in discussion, comparing previous studies)

#1. Why did the authors include only women populations?

#2. It would be better to summarize what the authors intended to mention in introduction section.

Reviewer #2: Dear editor,

The manuscript is a longitudinal study that aims to evaluate the impacts of health assessments on long-term care utilisation, physical functioning, and mortality in a cohort of older women aged 75+. The study showed that women who had an assessment had significantly lower odds of poor outcomes than women who had no assessment. However, women who had health assessments also increased healthcare services utilization, which is likely to be an important mechanism in improving the women’s health outcomes.

Major Comments

- There are many trials in the literature studying the effects of health assessments on older people’s several outcomes. The paper provides a good overview, but it would be interesting to state in the introduction how the paper adds to this branch of the literature. What are the lessons of this paper that are not previously found in the literature?

- The effects of health assessments appears to be strongly coincident of a greater use of health services. One possible extension to further understand the effects of health assessments would be to perform a dose-response exercise to check if a great number of health assessments is associated with positive outcomes.

- Some parts of the propensity score matching section would benefit from some rewriting. For instance, the beginning of the section reads as “participants who had never had an assessment were assigned into blocks based on the date of their first health assessment”. This sentence seems contradictory. In addition, it would also be important to describe the list of covariates, squared terms, and interaction terms in a supplementary material to facilitate the reproduction of the paper’s results.

- One big concern of propensity matching (discussed by the authors in page 19) is that residual confounders may bias the results. It would be interesting to perform an “out of the sample” exercise as follows: select a characteristic or a set of characteristics that was not used in the propensity score model and check if there are statistically significant difference between participants who did and did not have health assessment.

- The paper discusses that the matched sample is different from the original population. It would be interesting to know whether this difference might influence the key results of the paper.

Minor Comments

- The manuscript would benefit from a proofreading review for typos, which can be also found in the abstract.

- I suggest improving the Figure 2 quality.

- It seems the title describing Fig 3 has a comment in brackets that does not make sense. This comment might be misplaced, and it is likely that the comment refers to Fig 2.

- It is important to add that there are six survey rounds in the study.

- It was not clear in the “Outcome Variables Section” the years which healthcare utilization data are available

- It would be interesting to provide a brief intuition of the prognostic scores described on page 14. For instance, why does the score increase with age?

- On page 14, it seems that one of the ranges of the IQR has a typo.

- It seems that the results described in the last paragraph on page 16 are missing from Table 2.

6. PLOS authors have the option to publish the peer review history of their article (what does this mean?). If published, this will include your full peer review and any attached files.

Reviewer #1: No

Reviewer #2: No

---

## [Author Response · Author response to Decision Letter 0]

10 Feb 2021

Please refer to 'Response to Reviewers' word document.

---

## [Decision Letter · Decision Letter 1]

15 Mar 2021

Estimating the effect of health assessments on mortality, physical functioning and health care utilisation for women aged 75 years and older

PONE-D-20-34902R1

Dear Dr. Byles,

We’re pleased to inform you that your manuscript has been judged scientifically suitable for publication and will be formally accepted for publication once it meets all outstanding technical requirements.

Kind regards,

Y Zhan

Academic Editor

PLOS ONE

Additional Editor Comments (optional):

Reviewers' comments:

Reviewer's Responses to Questions

**Comments to the Author**

1. If the authors have adequately addressed your comments raised in a previous round of review and you feel that this manuscript is now acceptable for publication, you may indicate that here to bypass the “Comments to the Author” section, enter your conflict of interest statement in the “Confidential to Editor” section, and submit your "Accept" recommendation.

Reviewer #1: All comments have been addressed

Reviewer #2: (No Response)

2. Is the manuscript technically sound, and do the data support the conclusions?

Reviewer #1: Yes

Reviewer #2: Yes

3. Has the statistical analysis been performed appropriately and rigorously? 

Reviewer #1: Yes

Reviewer #2: Yes

4. Have the authors made all data underlying the findings in their manuscript fully available?

Reviewer #1: Yes

Reviewer #2: Yes

5. Is the manuscript presented in an intelligible fashion and written in standard English?

Reviewer #1: Yes

Reviewer #2: Yes

6. Review Comments to the Author

Reviewer #1: This manuscripted have been revised according to the reviewer's comments.

It seems appropriate for the publications.

Reviewer #2: In my opinion, the new version of the paper addresses my points. However, it seems the Fig 3 is missing from the manuscript. I suggest revising this point.

7. PLOS authors have the option to publish the peer review history of their article (what does this mean?). If published, this will include your full peer review and any attached files.

Reviewer #1: No

Reviewer #2: No

---

## [Editor Report · Acceptance letter]

25 Mar 2021

PONE-D-20-34902R1 

Estimating the effect of health assessments on mortality, physical functioning and health care utilisation for women aged 75 years and older 

Dear Dr. Byles:

I'm pleased to inform you that your manuscript has been deemed suitable for publication in PLOS ONE. Congratulations! Your manuscript is now with our production department. 

Kind regards, 

on behalf of

Dr. Y Zhan 

Academic Editor

PLOS ONE